REGISTERED REPORT PROTOCOL

# Assessment of effectiveness of DAMaN: A malaria intervention program initiated by Government of Odisha, India

**Madhusmita Bal[1], Arundhuti Das[1], Jyoti Ghosal[1], Madan Mohan Pradhan[2], Hemant Kumar Khuntia[1], Sanghamitra Pati[1]ᵒ, Ambarish Dutta[3]ᵒ, Manoranjan Ranjit[1]ᵒ ***

**1** ICMR- Regional Medical Research Centre, Bhubaneswar, Odisha, India, **2** State Vector Borne Disease Control Programme, Boudh, Odisha, India, **3** Indian Institute of Public Health, Bhubaneswar, Odisha, India

ᵒ These authors contributed equally to this work.

* ranjit62@gmail.com

## Abstract

India, a persistently significant contributor to the global malaria burden, rolled out several anti-malaria interventions at the national and state level to control and recently, to eliminate the disease. Odisha, the eastern Indian state with the highest malaria burden experienced substantial gains shown by various anti-malaria initiatives implemented under the National Vector-borne Disease Control Programme (NVBDCP). However, recalcitrant high-transmission "pockets" of malaria persist in hard-to-reach stretches of the state, characterised by limited access to routine malaria surveillance and the forested hilly topography favouring unbridled vector breeding. The prevalence of asymptomatic malaria in such pockets serves as perpetual malaria reservoir, thus hindering its elimination. Therefore, a project with the acronym DAMaN was initiated since 2017 by state NVBDCP, targeting locally identified high endemic 'pockets' in 23 districts. DAMaN comprised biennial mass screening and treatment, provisioning of long-lasting insecticidal net (LLIN) and behavioural change communication. Subsequently, to inform policy, assessment of DAMaN was conceived that aims to estimate the coverage of the various components of the project; the prevalence of malaria, even at sub-patent level especially among pregnant/lactating women and children; and its impact on malaria incidence. A survey of DAMaN beneficiaries will measure coverage; and knowledge and practices related to LLIN; along with collection of blood specimens from a probability sample. A multi-stage stratified clustered sample of 2228 households (~33% having pregnant/lactating women) will be selected from 6 DAMaN districts. Routine DAMaN project data (2017–2018) and NVBDCP data (2013–2018) will be extracted. Rapid Diagnostic Test, Polymerase Chain Reaction and blood smear microscopy will be conducted to detect malarial parasitemia. In addition to measuring DAMaN's coverage and malarial prevalence in DAMaN pockets, its impact will be estimated using pre-post differences and Interrupted Time Series analysis using 2017 as the "inflection" point. The assessment may help to validate the unique strategies employed by DAMaN.

**Data Availability Statement:** All relevant data from this study will be made available upon study completion.

**Funding:** The research grant was received by MR, MB, AD and SP. Funding received from: Department of Health and Family Welfare, Govt. Of Odisha (DHS:2244/DAMaN/Project proposal/18-19/31.12.2018) https://health.odisha.gov.in The funder will not have a role in study design, data collection and analysis, decision to publish, or preparation of the manuscript.

**Competing interests:** The authors have declared that no competing interests exist.

# Introduction

Malaria, the age-old mosquito borne parasitic disease, remains a substantial health problem in many parts of the world, especially in the tropical developing nations. The Millennium Development Goal-4 [1] followed by the Sustainable Development Goals -3 [2] of the United Nations trained their focus on malaria, as because it is a key contributor towards global burden of mortality and morbidity, especially among children under five years of age (under-5), a key segment of the population for the developmental goals. To attain this, World Health Organization (WHO) adopted the comprehensive Global Technical Strategy, with an aim to reduce malaria incidence and mortality rates by at least 90%, worldwide, by 2030 [3].

India is one of the 11 high malaria burden countries in the world. Along with Sub-Sahara African countries it contributes to 85% of global malaria burden, despite a sharp reduction (28%) of malaria cases from 2017 to 2018 in the country [4]. India has expansive geography and diverse climate; therefore, its certain regions provide an ideal environment for sustaining malaria parasites and their vectors [5]. Ambient temperature, relative humidity, forested hilly topography and extended rainy season of these regions favour malaria transmission. Eventually, in Odisha, an eastern coastal state of India, especially in its two geo-physical regions (the Eastern Ghats and Northern plateau), such conditions [6] abound. Therefore, it is not surprising that the endemicity of malaria in Odisha has always been significantly high historically [7–9] as it contributed to 40% of the countries' total burden of malaria in 2017 [10].

In India, many programmes have been rolled out nationwide to prevent and control malaria since independence. The National Malaria Control Program (NMCP) was launched in 1953 and delivered astounding results within a five-year period. Thereafter, NMCP was renamed to National Malaria Eradication Program (NMEP) in 1958 with a view to eliminating the disease entirely from the country. However, unfortunately, malaria staged a huge comeback in India as the anti-malaria resources were prematurely withdrawn after the initial remarkable successes of the '50s and '60s. This surge in malaria led to the launch of Modified Plan of Operation in 1977 with an eye towards reduction of the disease burden in a cost-effective and integrated manner. Time to time malaria action plan was further updated, as in 1995, when emphasis was given on the use of revised drug schedule in high-risk areas as effectivity of traditional anti-malaria drugs were on the wane. With further change in policy, the programme was renamed as National Anti-Malaria Programme (NAMP) in 1999. Later in 2002 the National Vector Borne Disease Control Program (NVBDCP) put other vector borne diseases of national concern along with malaria under one umbrella for optimum utilisation of manpower and resources. In 2005, an Intensified Malaria Control Project was launched with the assistance of Global Fund for AIDS, Tuberculosis and Malaria in North Eastern states and Odisha, Jharkhand and West Bengal [11]. It introduced new interventions for case management and vector control, namely Rapid Diagnostic Tests (RDT) (2005), Artemisinin-based Combination Therapy (ACT) (2006) and Long Lasting Insecticidal Nets (LLINs) (2009) [12]. These were then significantly reflected in the Strategic Plan for Malaria Control in India for 2012–2017. Robust methods of monitoring and evaluation were also incorporated into the programme, to track the new interventions [13].

Meanwhile, given the gravity of the problem, malaria always received priority in Odisha, which instituted anti-malaria initiatives in tandem with national measures. As mentioned above, after an initial reduction in reporting of malarial cases due to early success of NMCP, a resurgence of malaria was observed since 1967 in the state. After which many significant schemes were launched that helped to shore up the state public health machinery to fight against the disease, again in line with the renewed efforts instituted by the national government. Regardless of all these efforts, malaria endemicity in the state remained obstinately high.

A feature that consistently stood out is that malaria control in large swathes of Odisha is operationally difficult, as significant portion of the land has hilly terrain, forest cover with poor communication facilities along with left-wing extremism; and a large section of the residents of these hard-to-reach areas are indigenous tribal people [7], who often live in widely scattered small hamlets. This historically has been rendering many such malaria endemic villages poor accessibility to malaria services [14].

In 2007, the Odisha Health Sector Plan (OHSP) aided by the Department for International Development (DFID) of the United Kingdom was rolled out [15]. The components of the intensified anti-malaria strategies under OHSP included integrated vector control measures, mass use of RDT and ACTs for diagnosis and treatment, service decentralization (working through village-level Accredited Social Health Activist–ASHA), behaviour change communication (BCC), improved surveillance, and inter-sectorial convergence. The "Mo Masari", meaning "our bed net" scheme of OHSP was also rolled-out to promote the use of LLINs among the pregnant women and tribal residential school children in 7 high malaria endemic districts of the state [16]. In tribal villages traditional folk-art based methods were used for knowledge dissemination. Subsequently during 2008 to 2013 these intensified anti- malarial activities in the state helped to reduce > 44% notification of malaria [15]. However, this decline could not be sustained in the consecutive years (Fig 1) especially in the districts having high forest cover and many hard-to-reach areas.

In 2013, the Comprehensive Case Management Project (CCMP), an operational research initiative, was started by NVBDCP, Odisha in collaboration with National Institute of Malaria Research, with support from Medicines for Malaria Venture and WHO [14]. CCMP showed that high endemic hard-to- reach villages had sub-optimal malaria surveillance, often displaying misleadingly low reporting of cases due to under-detection, and thus ultimately leading to persistence of malaria reservoir in those pockets. Majority of such persistent cases were found to be afebrile or asymptomatic, therefore not seeking routine malaria care, making them even more critical to local malaria transmission as they would not be identified by routine malaria surveillance system. CCMP also demonstrated that mass screening and treatment in this hard-to-reach, forest covered areas can be remarkably useful in detection and treatment of such asymptomatic/ afebrile malaria cases. Another similar study conducted in hard-to-reach areas of southern districts of Odisha went on to illustrate that detecting and treating afebrile malaria

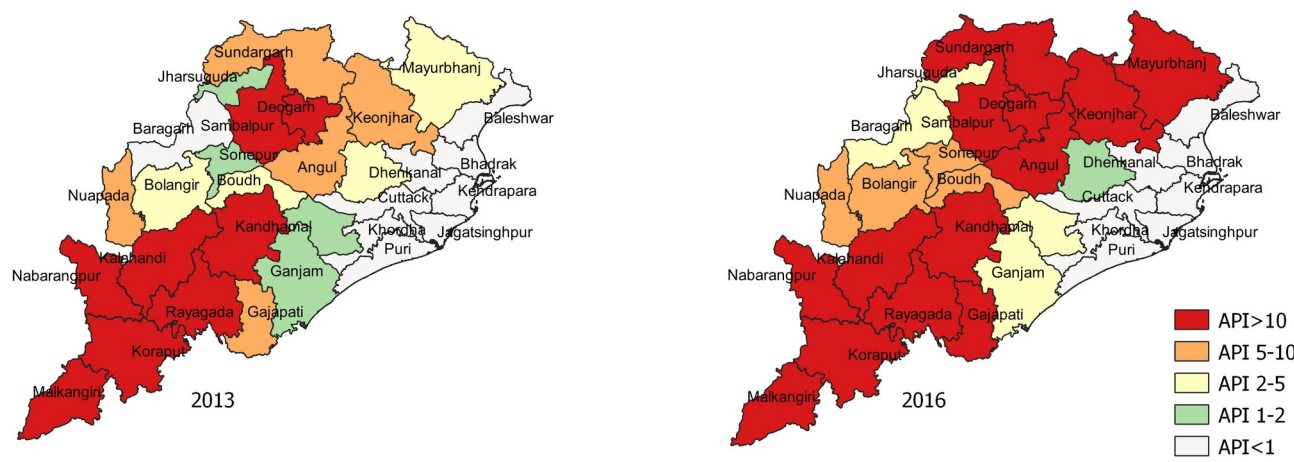

**Fig 1. Odisha Annual Parasite Incidence (API) status of the year 2013 and 2016.** The data originated from Balk, D., M. R. Montgomery, H. Engin, N. Lin, E. Major and B. Jones. 2020. Spatial Data from the 2011 India Census. Palisades, NY: NASA Socioeconomic Data and Applications Center (SEDAC). https://doi.org/10.7927/gya1-wp91. Accessed 29 July 2020. No copyrighted material was used. The data was further modified using QGIS software version 3.4.13.

in under-5 children in hard-to-reach areas improved growth, anaemia and malaria case load [17].

From these observations it became imperative that in order to achieve malaria elimination in Odisha, asymptomatic malaria cases residing in these hard-to-reach areas of the state have to be identified and treated in order to contract the persisting malarial reservoirs. Meanwhile, the usefulness of LLIN from previous experience in the state also led planners to include this tool in a comprehensive package, which was then conceived by the state government as a large-scale innovative project called "Durgama Anchalare Malaria Nirakarana (DAMaN)" (English translation: "Malaria Elimination in Remote Areas") in 2017. The main objective of DAMaN was to supplement the routine malaria control activities and fill the gaps in case finding and treatment as well as vector control in the high-endemic remote pockets of the state. In 2017, this massive public health initiative was rolled out in the hard-to-reach pockets of 23 high malaria burden districts of the state, targeting roughly a population of 1 million residing in 7000 far-flung villages. The three pillars of the DAMaN strategy included 1) mass screening (screening of all residents) by using bivalent RDT followed by treatment of RDT-positive cases in a "camp approach"—referred to as Mass Screening and Treatment (MSAT), 2) vector control measures that included LLIN distribution and promotion, and lastly 3) community mobilisation. Table 1 summarises the aims and approaches of DAMaN.

The MSAT operations of DAMaN were planned to be carried out mainly through ASHA and village-level community volunteers twice a year under the supervision of health workers and health supervisors, first during pre- monsoon (March-April-May) when the entire population in the target villages/hamlets were to be covered and second during post-monsoon season (October-November) when only pregnant women and children from the target villages/hamlets were to be screened and treated.

DAMaN is a massive new public health initiative to control malaria in high endemic and hard-to-reach underserved areas. It entailed an investment of 10 crore Indian Rupees (1333591 US dollars) per year, which is a significant amount for a relatively resource-constrained state like Odisha. Early results show that implementation of DAMaN coincided with drastic reduction of case notification in high endemic districts [10]. Therefore, an exercise was conceptualised to comprehensively assess the DAMaN project (hereinafter referred to as *DAMaN Assessment*); its effectiveness and suggest mid-course corrections if found necessary. *DAMaN Assessment* is also of immense significance as it would categorically evaluate and validate the effectiveness of the various components of DAMaN, so that its successful pieces can be further replicated in other malaria endemic region of the country or around the globe.

**Table 1. Aims and approaches of DAMaN.**

| Aims of DAMaN | Approach | Target population | Identification of target population |
|---|---|---|---|
| To reduce the malaria parasite load including number of gametocytes in the human population, especially among asymptomatic/afebrile cases | MSAT in camp approach twice a year | Underserved population in hard-to-reach villages/hamlets, many of whom are asymptomatic/afebrile | Through local NVBDCP network who have local knowledge and based on historic NVBDCP case notification |
| To kill the anopheles vector mosquitoes and to provide personal protection to the villagers from mosquito bites so as to break the "human-mosquito-human" cycle | LLIN distribution and promotion | Population in hard-to-reach areas and also otherwise high endemic "pockets". Entire population of all villages was protected by LLIN. For this each household was given adequate numbers of LLIN depending upon the number of people in the house hold (one LLIN per 1.8 People). | NVBDCP case notification rates |
| To mobilize the community and carry out BCC for DAMaN so that the target population utilizes the service packet of DAMaN | Village contact drive, school sensitization and Malaria Shamadhan Sivira (MSS) | Underserved population in hard-to-reach villages/hamlets | Local NVBDCP network |

The main objectives of the *DAMaN Assessment* include

Estimation of the coverage of the three main components of DAMaN that includes mass screening and treatment (MSAT), vector control measures (LLIN and/or IRS) and community mobilisation and participation.

Estimation of the serial prevalence of malaria parasites at clinical and sub-clinical (asymptomatic and sub-patent) level of parasitemia.

Evaluation of the impact of DAMaN on fever and malaria burden as reflected in NVBDCP data.

Assessment of the prevalence of maternal and child health in terms of birth outcome, anaemia in pregnancy and nutritional status of under five children among DAMaN beneficiaries and compare with state averages.

## Methodology

### Approaches and design of the *DAMaN Assessment*

Table 2 summarises the various objectives of *DAMaN Assessment* and the approaches to address each objective.

### Ethical approval

The study has been approved by both the Institute Human Ethical Committee (ICMR-RMRC/ IHEC-2019/012 dated 27/02/2019) and Research & Ethics Committee of Department of Health and Family Welfare, Govt. Of Odisha (453 /SHRMU /187 /17 Dated 22/8/2017).

**Table 2. The objective(s), approaches and study design of the *DAMaN Assessment*.**

| Objectives | Approaches and study design |
|---|---|
| 1. To estimate the coverage of the three main components of DAMaN that includes | 1a. Coverage of MSAT, a component of DAMaN will be estimated along with MSAT's test positivity rate proportion of asymptomatic among positives etc. This will be done from DAMaN project information system data. |
| a) mass screening and treatment (MSAT), | |
| b) vector control measures (LLIN) and c) community mobilisation and participation | 1b. A survey of DAMaN beneficiaries to be conducted (details below) and primary data from the survey will be used to estimate Long lasting Insecticidal Net (LLIN) usage and coverage of Indoor Residual Spray |
| | 1c. Same data as 1b will be used to estimate coverage by DAMaN-related community mobilisation campaigns |
| 2. To estimate the prevalence of malaria parasites at clinical and sub-clinical (asymptomatic and sub-patent) level | 2a. Secondary data, related to MSAT, collected from DAMaN project information system for 2017 to 2019. This will help to track the trend of prevalence in parasitemia |
| | 2b. DAMaN survey will include collection of blood specimens from the sampled individuals and tests to identify parasitemia and quantification of parasite gametes will be carried out. Two rounds of the survey is to be conducted which will help us to track the trend in parasitemia prevalence |
| 3. To evaluate the impact of DAMaN on fever and malaria burden as reflected in routine NVBDCP data | Routine yearly and monthly NVBDCP data, collected from the level of the Sub-Centres, the lowest reporting units of the programme, for blood examination rate and parasite incidence for the six survey districts from 2013 to 2018 (6 years) will be analysed using appropriate variants of Difference analysis, Difference analysis across three categories of Sub-Centre stratified by the scale of DAMaN's MSAT coverage, *interrupted time-series* (ITS), ITS analysis across three strata of sub-center. The sub-centres then will be stratified by their DAMaN project's MSAT coverage data (see point 1). Then trends of NVBDCP data will be compared across the strata of Sub-Centres (hypothesis: "higher the coverage of DAMaN, steeper is the decline of malaria and fever incidence") |
| 4. To assess the maternal and child health in terms of birth outcome, anaemia in pregnancy and nutritional status of under five children among DAMaN beneficiaries and compare with state averages | A survey of DAMaN beneficiaries will be conducted and primary anthropometric, haemoglobin and pregnancy outcome indicators will be estimated. It will be compared with state averages of Odisha |

DAMaN project data will be extracted in aggregate form and no individual-level data will be retrieved or used, therefore there is no need for de-identification of DAMaN project data. There is also no plan to access patient medical records. Survey data and blood specimen will be collected after written consent/assent is taken from each participant. The raw survey data will be under the secure custody of the principal investigator and the data will be completely de-identified for analysis.

## Survey

A survey of DAMaN beneficiaries and their households is to be carried out to collect primary data on these individuals, which will also include collection of their blood specimens. A probability representative sample of the DAMaN beneficiaries will be drawn for the survey the calculation of the size and sampling technique of which are described below. The survey will be conducted from the month of August to November, which synchronies with the period of DAMaN implementation.

**Sample size.**  The sample size of households (each containing pregnant/lactating women) for survey has been estimated using the formula, n = [DEFF*Np(1-p)]/ [(d2/Z21-α/2*(N-1) +p*(1-p)], where the rarest event that needs to be examined in this survey is parasitemia in pregnant/lactating women and in areas with relatively low endemicity among DAMaN-covered villages it is assumed to be around 3%. Additionally, 1% absolute precision, Z score corresponding to 95% Confidence Interval that is 1.96, with design effect 2 have been applied to this calculation. Therefore, the sample size of households to be surveyed is 2228. Only the household head (also pregnant and lactating mother) will be interviewed per household. However, blood specimens will be collected from all members in the household present at the time of the interview.

The sample size calculation was based on two assumptions. The first being the prevalence of parasitemia among pregnant/lactating women in DAMAN areas being 3% and the second, pregnant women making up 1.5–2% of the population and 0–59 months old children making up approximately 10%. Hence, the prevalence of other more common events in other sub-population such as under-5s can be estimated using this sample conveniently.

**Sampling strategy.**  The broad sampling strategy to be followed is multi-stage clustered sampling technique. The various levels of sample elements and their clustered nature, relevant for this study can be seen in figure below (Fig 2).

## Data

Four types of data will be used for *DAMaN Assessment* which are as described below. A pre-tested, validated questionnaire inspired by that used in National Family Health Survey fourth round, 2015–2016 (NFHS-4) is to be applied to capture household and individual-level data, which is to be answered by the household head. The questionnaire consists of many sections, namely, the demographic and household information, the socio-economic characteristics and knowledge and practices of general malaria, vector control and treatment of fever cases. The questionnaire also will contain information on pregnant and lactating mother. At least 1 ml of intra-venous blood specimen will be collected for malaria and haemoglobin analysis and anthropometric measurements will be recorded from all the members of the sampled household present at the time of the interview. Pregnant women sampled in the survey will be followed-up longitudinally till their birth outcome can be recorded. The variables of the different survey components are listed in the Table 3.

**DAMaN project information system data.**  DAMaN project data will be extracted from the six sampled districts for the years 2017 to 2019. This will contain village-wise data of

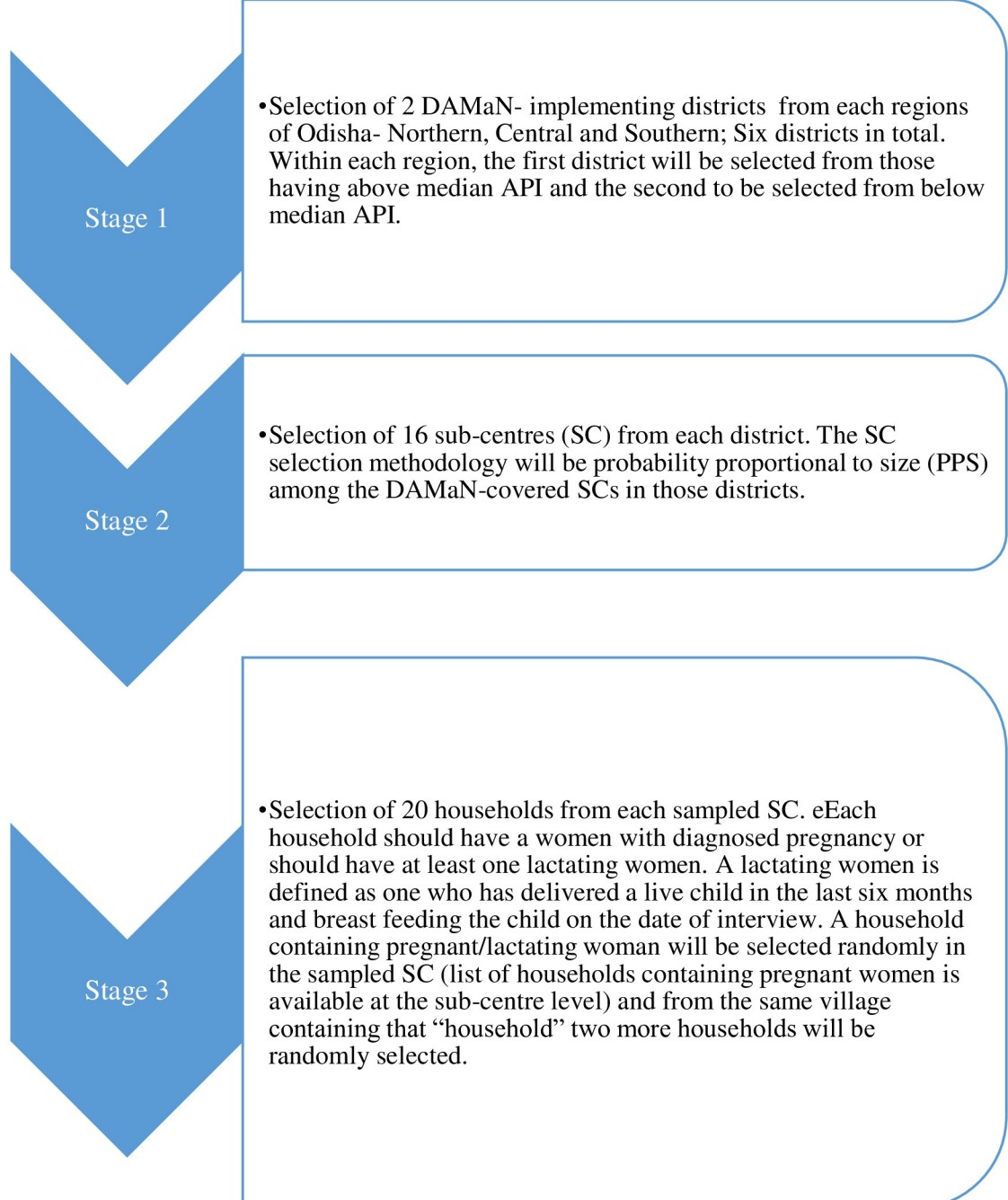

**Stage 1**

• Selection of 2 DAMaN- implementing districts from each regions of Odisha- Northern, Central and Southern; Six districts in total. Within each region, the first district will be selected from those having above median API and the second to be selected from below median API.

**Stage 2**

• Selection of 16 sub-centres (SC) from each district. The SC selection methodology will be probability proportional to size (PPS) among the DAMaN-covered SCs in those districts.

**Stage 3**

• Selection of 20 households from each sampled SC. eEach household should have a women with diagnosed pregnancy or should have at least one lactating women. A lactating women is defined as one who has delivered a live child in the last six months and breast feeding the child on the date of interview. A household containing pregnant/lactating woman will be selected randomly in the sampled SC (list of households containing pregnant women is available at the sub-centre level) and from the same village containing that "household" two more households will be randomly selected.

**Fig 2. The various levels of sample elements and their clustered nature data.**

MSAT carried out by DAMaN. Among positive cases whether they are symptomatic (Y/N) and the type of parasite(s) (*P falciparum or P Vivax* or both) will also be noted in addition to their demographic (age, sex and other) data.

**NVBDCP routine data.** Both yearly and monthly Sub-Centre (SC—the most peripheral health outpost in the Indian health system) wise routine NVBDCP data will be collected from the six sampled districts of Odisha that would contain parasite incidence and blood

**Table 3. Summary of the survey variables.**

| Survey Components | Types of tool | Variables |
|---|---|---|
| **Section I:** Demographic and household information | 12 questions | Demographic characteristics of all the sample household members will be collected in this section. Information on household members covered in malaria screening conducted under DAMaN project will also be recorded. |
| **Section II:** Anthropometry | 3 measurements | Height, weight, Mid-Upper Arm Circumference (MUAC) of the household members will be measured. |
| **Section III:** Socio-economic characteristics | 13 questions | Information on the assets owned by the beneficiary will be asked in order to analyse their socio-economic status of the household. |
| **Section IV:** Knowledge and awareness of malaria | 15 questions | This will include questions on awareness and knowledge of malaria, along with questions on health care seeking practices related to malaria. |
| **Section V:** Vector control | 10 questions | Questions related to vector control measures taken through the DAMaN project will be asked along with the usage pattern of mosquito nets |
| **Section VI:** Fever cases | 2 questions | Details of the family member who had fever at any time in the two weeks preceding the survey will be noted in this section. |
| **Section VII:** Information on pregnant and lactating mothers | 8 questions | Data regarding pregnancy history and birth outcome along with birth weights will be gathered in this section. |
| **Section VIII:** Blood Marker | 4 tests on blood specimens of all survey participants | Rapid Diagnostic Test (RDT), Polymerase Chain Reaction (PCR), Microscopy and Complete Blood Count (CBC) will be done with the blood specimen collected from surveyed population. (details below) |

examination rate, disaggregated by age, sex and type. The data of 7 years (2013 to 2019) will be collected and the period 2013–2017 will be considered pre-DAMaN phase and 2018–2019 post-DAMaN.

**National Family Heath Survey 4 (NFHS-4) factsheet.** The Odisha averages of Hb/anaemia, birth outcomes and anthropometry in the general population, pregnant/lactating women and children of Odisha will be extracted from the state-wise factsheet of NFHS-4 that was conducted in 2015–2016 [18].

## Laboratory investigation

The laboratory analysis will consist of four tests of the blood specimen collected from surveyed participants.

Three diagnostic methods, namely Rapid Diagnostic Test (RDT), blood smear microscopy, polymerase chain reaction (PCR) will be used to find the prevalence of malarial parasites in the blood specimen collected. This *modus operandi* will ensure an exhaustive investigation for detecting parasitemia. The Histidine-Rich Protein 2 (PfHRP2) gene deletion in *Plasmodium falciparum* leads to no-detection in RDT, however Pf prevalence in those cases can be captured using microscopy and PCR. The PCR method will also be helpful in detecting parasite at submicroscopic level, which usually gets missed out by RDT and microscopy.

**Rapid Diagnostic Test (RDT).** Primary screening of the malaria infection will be performed by using RDT (Pf/PAN malaria antigen) and microscopy followed by PCR for confirmation and species identification. The RDTs to be used in our study detects Histidine-Rich Protein 2 (PfHRP2) from *P.falciparum* and Parasite-Specific Lactate Dehydrogenase (pLDH) from the parasite glycolytic pathway found in all species. The RDT will assist in finding *P. falciparum* infection and/or a mixed infection. Mixed infection implies presence of more than one species of *Plasmodium* in the individual. The sensitivity and specificity of the kit is 98.72% and 96.76% respectively.

**Blood smear microscopy.** Both thick and thin blood smears will be prepared for diagnosis and species identification of malaria parasites within 24 hours of blood collection. Briefly blood smears will be fixed with methanol (thin smear only) and stained with Giemsa stain.

Smears will be examined microscopically and graded for the presence/absence and species of *Plasmodium*.

**Polymerase Chain Reaction (PCR).** To confirm ICT and/or microscopy results and detect parasites below the limit of microscopy (around 40 parasites/ml of blood) a retrospective, species-specific PCR will be carried from each blood samples. The DNA will be extracted from the blood sample using Qiagen DNA mini kit (Quigen,West Sussex,UK). Separate reactions will be carried out using species-specific (*P.falciparum*, *P.vivax*, *P.ovale*, *P. malariae*) oligonucleotide primers with every sample for the detection of each species in a reaction volume of 20µl as described by others [19] and regularly performed in our laboratory for routine malaria diagnosis. The PCR amplified products will be separated by electrophoresis on 2% agarose gel with added ethidium bromide (0.5µg/ml). Gels will be visualised and processed using gel documentation system.

**Haematological analysis.** Haemoglobin estimation as a part of the complete blood cell count (CBC) will be done in automatic haematological analyser (Melet Schloesing Laboratories, USA) within 24 hour of collection of whole blood in EDTA.

## Statistical analysis

**Descriptive statistics.** Age and education will be categorized from quantitative variables. Body Mass Index (BMI) (weigh in kg/height in m2) will be categorized into underweight (<18.5 kg/m2), normal (18.5–24.9 kg/m2) and overweight/obese (> 25 Kg/m2). Stunting (height for age, Wasting (weight for height) and underweight (weight for age) will be defined as mild/moderate, or severe when their values are < -2 Standard Deviation (SD) and < -3SD respectively from the median value of the WHO Child Growth Standards for under-five children. The household-level wealth index is to be based on the recorded assets and housing and sanitation facilities. Principal component analysis will be used to derive a single wealth index from these multiple variables, which will be then used as a quantitative variable and also categorized into five quintiles to be used as an ordinal variable. Anaemia will be defined using WHO (2001) prescribed cut-off values (Table 4).

Summary statistics will be used to describe the various features of primary study sample and prevalence of anaemia, various measures of undernutrition and prevalence of malaria infection in the sample population. The descriptive statistics will also summarise LLIN coverage and usage, IRS activities and malaria-related knowledge and practices in the sampled DAMaN villages. The relevant indicators from DAMaN survey data will be compared with that of Odisha state averages extracted from NFHS-4 factsheet [18].

Statistics such as test coverage, test-positivity and asymptomatic among positives will be used to summarise DAMaN project data, which will be aggregated at the sub-district (block) level for the six sampled districts of *DAMaN Assessment*.

**Table 4. Haemoglobin cut-off values used to define anaemia.**

| Reference group | Non-Anaemia (g/dl) | Categories of Anaemia (g/dl) | | |
|---|---|---|---|---|
| | | Mild | Moderate | Severe |
| Children 6–59 months of age | 11 & above | 10–10.9 | 7–9.9 | <7 |
| Children 5–11 years of age | 11.5 & above | 11–11.4 | 8–10.9 | <8 |
| Children 12–14 years of age | 12 & above | 11–11.9 | 8–10.9 | <8 |
| Non-pregnant women (15 years of age and above) | 12 & above | 11–11.9 | 8–10.9 | <8 |
| Pregnant women | 11 & above | 9–10.9 | 7–9.9 | <7 |
| Men (15 years of age and above) | 13 & above | 11–12.9 | 8–10.9 | <8 |

**Difference analysis.** The difference in average annual case notification (also known as Annual Parasite Incidence) between post-DAMaN (2018–2019) and pre- DAMaN (2013–2017) will be computed using a Poisson regression framework as positive malaria cases notified is count data which approximates a Poisson distribution. The equation has been explained below.

$$\log(y) = \beta_0 + \beta_1 x + \text{offset}(\log(\text{population})) + \varepsilon \qquad (1)$$

$y$ = annual count of malaria cases notified

x = phase expressed as 0, 1; where 0 indicates pre-DAMaN and 1 indicates post-DAMaN.

$\beta_1$ is the parameter of interest—the exponentiated value of which will give the ratio of average annual case notification in post-DAMaN: pre-DAMaN phases.

**Difference analysis across three categories of sub-centre stratified by the scale of DAMaN coverage.** The initial equation to test whether the ratio of average annual case notification in post-DAMaN: pre-DAMaN phases vary across the three categories of sub-centre significantly is as follows:

$$\log(y) = \beta_0 + \beta_1 x_1 + \beta_2 x_2 + \beta_3 x_1 * x_2 + \text{offset}(\log(\text{population})) + \varepsilon \qquad (2)$$

Where:

$y$ = annual count of malaria cases notified

$x_1$ = time period / intervention phase expressed as 0, 1; where 0 indicates phase 1 and 1 indicates phase 2.

$x_2$ = a categorical variable indicating three strata of sub-centres (0 = no DAMaN coverage, 1 = below median coverage, 2 = above median coverage)

$x_1*x_2$ = indicates the interaction between phase and the intervention status.

Log(population) of the SC will be used as the offset variable in the equation.

Greek letters $\beta_0$, $\beta_1$, $\beta_2$ and $\beta_3$ are all unknown parameters to be estimated and $\varepsilon$ is a random, unobserved 'error' term.

The coefficients:

$\beta_3$ the focus is whether this parameter is significant or not.

If found significant then using Eq (1) annual case notification ratios will be estimated for three strata of sub-centres separately to observe whether coverage by DAMaN (the stratifying variable) is related to decline in cases.

**Interrupted time-series analysis.** Sub-centre-wise monthly time-series data, of cases notified and blood specimens examined routinely by NVBDCP, will be dealt with interrupted time-series analysis approach that will use a segmented Poisson regression framework. The count outcome data will be regressed over time to estimate the "trend" of these indicators. The trend will be "segmented" that is divided by an "inflection" (interruption) point, which corresponds to 2018 after DAMaN was rolled out in 2017. Immediate changes in "level" of the trend lines across the inflection point will also be estimated.

The segmented regression equation to be used in the ITS analysis will be:

$$\log(y) = \beta_0 + \beta_1 x_1 + \beta_2 x_2 + \beta_3 x_1 * x_2 + \text{offset}(\log(\text{population})) + \varepsilon \qquad (3)$$

Where:

$y$ = count of outcome indicator

$x_1$ = time since the start of the study

$x_2$ = phase expressed as 0, 1; where 0 indicates phase 1 and 1 indicates phase 2.

$x_1*x_2$ = represent interaction term

$\beta_1$ = the slope of pre-intervention.

$\beta_2$ = exponentiated value represents the change in the "level" (immediate change) following the introduction of the intervention expressed as ratio of immediate shift in monthly case detection post-DAMaN: pre-DAMaN

$\beta_3$ = exponentiated value represents the difference between slopes between two phases of intervention (long-term change) expressed as ratio of trends in monthly case detection post-DAMaN: pre-DAMaN

**Interrupted time-series analysis across three strata of sub-centre.** We will test whether the post-DAMaN: pre-DAMaN "slope" or "level" changes are significantly different across three categories of sub-centre using a segmented Poisson regression for which the equation is following:

$$\log(y) = \beta_0 + \beta_1 x_1 + \beta_2 x_2 + \beta_3 x_3 + \beta_4 x_1 * x_2 + \beta_5 x_2 * x_3 + \beta_6 x_1 * x_3 + \beta_7 x_1 * x_2 * x_3 \\ + \text{offset}(\log(\text{population})) + \varepsilon \qquad (4)$$

Where:

$y$ = count of outcome indicator

$x_1$ = time as a continuous variable since the start of the data collection

$x_2$ = phase expressed as 0, 1; where 0 indicates phase 1 and 1 indicates phase 2.

$x_3$ = a categorical variable indicating three strata of sub-centres

$x_1{}^*x_2$; $x_2{}^*x_3$; $x_1{}^*x_3$ = indicates two-way interactions respectively

$x_1{}^*x_2{}^*x_3$ = three-way interaction between time, phase and intervention status

Log(population) of the SC will be used as the offset variable in the equation.

The coefficients of interest in ITS:

$\beta_6$ = is the first parameter of interest whether the variations in "level" changes are statistically significant or not across the three strata of sub-centres

$\beta_7$ = is the second parameter of interest whether the variations in "slope" changes are statistically significant or not across the three strata of sub-centres

If these parameters are significant then stratified interrupted time-series analysis using segmented Poisson regression (Eq 3) will be carried out in three strata of sub-centres separately.

All these 4 types of model will also be repeated with "blood examination" as the outcome.

R statistical software version 3.5.1 [20] will be used for all analyses. Specifically, WHO anthro package for R will be used to calculate the anthropometric measurement (stunting, wasting and underweight) for under five children.

## Discussion

The DAMaN intervention project is the first-of-its kind in the country targeting malaria elimination in hard-to-reach "pockets" of high malaria-burden district of the state by Government of Odisha. The proposed *DAMaN Assessment* will provide evidence on the impact and outcome of intervention based on reduction in overall parasite load and reduction/interruption in transmission and changes in important key health indicators. The study will also capture information on the maternal and child health as evidences exist that prolonged exposure to malaria has impacts on maternal health, pregnancy outcome, nutritional status and anaemia level of children under the age of five [21, 22]. Molecular investigation, another integral component of *DAMaN Assessment*, will determine not only the distribution of different species of *Plasmodium* in these areas even at sub patent level but also elucidate the changes of their behaviour as well as adaptation (if any) due to rigorous drug pressure as shown previously [23]. The extent of DAMaN coverage will help the programme managers of the project to realize the immediate output and course correction measures required, wherever necessary for their efforts, and in the long run will help to predict the ideal coverage required in order to arrest the transmission

of malaria in Odisha. Overall *DAMaN Assessment* will validate the measures implemented to eliminate malaria in these high burden remote areas of the state. The results of the study will inspire replication of the project in other parts of the country with similar malaria burden and topographic hindrance. The methodology emanating from this study could also be applicable in assessing the impact of other national/state projects related to elimination of other vector-borne diseases.

## Supporting information

**S1 File. Survey questionnaire.**
(PDF)

## Acknowledgments

We would like to thank Dr. Prameela Baral, Addl. Director of Health Services (VBD), Odisha, Dr P K Sahu, Nodal Officer NVBDCP-Odisha, Dr Kirti Mishra, Mr Debakanta Sandhibigraha and administration of the Department of Public Health, Odisha for supporting us with information about the processes and data sources of the DAMaN intervention project, which will be used for the assessment exercise in future.

## Author Contributions

**Conceptualization:** Madhusmita Bal, Sanghamitra Pati, Ambarish Dutta, Manoranjan Ranjit.

**Data curation:** Arundhuti Das, Jyoti Ghosal, Ambarish Dutta.

**Formal analysis:** Arundhuti Das, Jyoti Ghosal, Ambarish Dutta.

**Funding acquisition:** Madhusmita Bal, Sanghamitra Pati, Manoranjan Ranjit.

**Investigation:** Arundhuti Das, Hemant Kumar Khuntia, Manoranjan Ranjit.

**Methodology:** Madhusmita Bal, Jyoti Ghosal, Hemant Kumar Khuntia, Ambarish Dutta, Manoranjan Ranjit.

**Project administration:** Madhusmita Bal, Madan Mohan Pradhan, Sanghamitra Pati, Manoranjan Ranjit.

**Resources:** Madan Mohan Pradhan.

**Supervision:** Sanghamitra Pati, Manoranjan Ranjit.

**Writing – original draft:** Madhusmita Bal, Arundhuti Das.

**Writing – review & editing:** Ambarish Dutta, Manoranjan Ranjit.

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
