## [Decision Letter · Decision Letter 0]

12 Jun 2020

PONE-D-20-12484

Assessment of effectiveness of DAMaN : A malaria intervention program initiated by Government of Odisha, India

PLOS ONE

Dear Dr. Ranjit,

Thank you for submitting your manuscript for review to PLoS ONE. After careful consideration, we feel that your manuscript will likely be suitable for publication if the authors revise it to address specific points raised by the reviewer.  According to reviewers, there are some specific areas where further improvements would be of substantial benefit to the readers, including sample size calculation. Additionally, the authors should clarify about critical variables that may impact malaria control, such as the annual variation in rainfall and the fluctuation of vector density.

Please submit your revised manuscript by July 10.  If you will need more time than this to complete your revisions, please reply to this message or contact the journal office at plosone@plos.org. Please include the following items when submitting your revised manuscript:

We look forward to receiving your revised manuscript.

Kind regards,

Luzia Helena Carvalho, Ph.D.

Academic Editor

PLOS ONE

Journal Requirements:

2.  Please include additional information regarding the survey or questionnaire used in the study and ensure that you have provided sufficient details that others could replicate the analyses. For instance, if you developed a questionnaire as part of this study and it is not under a copyright more restrictive than CC-BY, please include a copy, in both the original language and English, as Supporting Information."

3. In your Methods section, please provide additional information about the participant recruitment method and the demographic details of your participants. Please ensure you have provided sufficient details to replicate the analyses such as: a) the recruitment date range (month and year), b) a description of any inclusion/exclusion criteria applied to participant recruitment, c)  descriptions of where participants will be recruited and where the research will take place. Moreover, in ethics statement in the manuscript and in the online submission form, please provide additional information about the patient records (DAMan project data) that will be used in the  retrospective part of study. Specifically, please ensure that you have discussed whether all data will be fully anonymized before you accessed them and/or whether the IRB or ethics committee waived the requirement for informed consent. If patients provided informed written consent to have data from their medical records used in research, please include this information.'

4. PLOS ONE does not permit references to unpublished data; therefore, we request that you either include the referenced data or remove the instances of "data not shown," "unpublished results," or similar.

5. We suggest you thoroughly copyedit your manuscript for language usage, spelling, and grammar. If you do not know anyone who can help you do this, you may wish to consider employing a professional scientific editing service.  

Reviewers' comments:

Reviewer's Responses to Questions

**Comments to the Author**

1. Does the manuscript provide a valid rationale for the proposed study, with clearly identified and justified research questions?

Reviewer #1: Yes

Reviewer #2: Yes

2. Is the protocol technically sound and planned in a manner that will lead to a meaningful outcome and allow testing the stated hypotheses?

Reviewer #1: Yes

Reviewer #2: Yes

3. Is the methodology feasible and described in sufficient detail to allow the work to be replicable?

Reviewer #1: Yes

Reviewer #2: Yes

4. Have the authors described where all data underlying the findings will be made available when the study is complete?

Reviewer #1: Yes

Reviewer #2: No

5. Is the manuscript presented in an intelligible fashion and written in standard English?

Reviewer #1: Yes

Reviewer #2: Yes

6. Review Comments to the Author

You may also provide optional suggestions and comments to authors that they might find helpful in planning their study.

Reviewer #1: PONE-D-20-12484

Title: Assessment of effectiveness of DAMaN: A malaria intervention program initiated by Government of Odisha, India

Article Type: Registered Report Protocol

General Comments: This study aims to assess the impact of a special programme launched in Odisha state of India to flush out asymptomatic malaria by biannual mass surveillance programme and of the mass distribution of long lasting insecticide nets (LLINs) on overall malaria incidence in the state. For the same, authors’ have proposed to divide malarious districts in to three strata based on 2016 malaria data (figure 1).

Specific Comments:

1. Figure 2 suggests that two districts are already selected from each of the three strata. Selection of the study districts in each stratum should be unbiased for which, the districts in each stratum may be listed from higher to lower API and then divided further in two subgroups of almost equal number of districts, one districts from upper half and one from the lower half in each zone may be selected. Sample size should be calculated based on lowest prevalence. This sample size may be applied to higher prevalence district in each of the zones for uniformity. Since prevalence of malaria has shown significant reduction in the years 2018 and 2019 in Odisha, sample size calculation should be based on current prevalence/incidence for representativeness and not based on 2016 data.

2. Introduction: Authors have stated ‘This initiative saw huge investment of public fund from the state government.’ How much per capita was invested, may be mentioned?

3. “At least 1 ml of intra- venous blood specimen will be collected for malaria and haemoglobin analysis and anthropometric measurements will be recorded from all the members of the sampled household present at the time of interview.”

Considering only two years of implementation of DAMaN, what is the rationale of taking anthropometric measurements? What difference DAMaN would have made in a short span to these features, is not understood? Hence this objective may be dropped and focus should be primarily on the epidemiological impact.

4. What definition of an asymptomatic case will be used? Will temperature be taken at the time of house visit?

5. Where is P. malariae in the whole scheme? This species has been reported from Odisha with decent prevalence in some districts as in previous studies. [The prevalence of P. malariae in Odisha, India by Pati et. al., Tropical Biomedicine 34(3): 607–614 (2017)]

6. The rationale for using three diagnostic methods may be clearly stated in view of Pfhrp-2 gene deletion which ranges from 5-10%; e.g. what is missed by RDT could be captured by microscopy and the PCR will capture low/sub-microscopic parasitaemia which the above two methods may miss detection.

7. On page 15, objective (d), Remove `prevalence’ to assess the outcome of maternal and child health.

8. On page 15, explain outcome assessment from MSAT? (Use Primary data/Generate data)

9. On page 16, Aim 3: Explain about stratification in detail in methodology.

10. On page 17, Sample size: Based on the calculated sample size, you should include 2228 household with pregnant women not household only.

11. On page 22, Statistical analysis: Anaemia should be defined separately for pregnant women, children and general population in a tabular form. It is not clear.

12. Authors will need to calculate sample size separately for LLIN usage (No. of Household) and child health outcome (<5 years)

Reviewer #2: Since 2017, the Indian National Vector-borne Disease Control Programme (NVBDCP) has implemented an intervention project (called DAMaN) to eliminate malaria in the hard-to-reach areas of high malaria-burden district of Odisha State, located on the south-eastern coast of the country.

Madhusmita Bal and colleagues are now proposing an evaluation study of the DAMaN project, in order to establish a continuous assessment of its effectiveness. This proposed DAMaN Assessment will provide evidence on the impact and outcome of intervention based on reduction in overall parasite load and reduction/interruption in transmission and changes in important key health indicators.

The strategies used in the DAMaN project to eliminate malaria in the target areas are not innovative. All of them were based on WHO guidelines for the elimination of malaria transmission. However, the evaluation methodology proposed by the authors of this manuscript brings a new tool to the malaria control and elimination strategies.

The rationale of this registered report protocol is clear, relevant and valid. The study protocol is well written and the methodology stated the authors hypotheses. The sample size was properly calculated. The strategies to analyse the results of the DAMaN Assessment study consider proposals for adequate, robust and well-characterized statistical models.

Despite the clarity of the study protocol submitted, it was not evident how the authors will analyse some variables provided for in the DAMaN Project. In the main objectives of the DAMaN Assessment the authors stated that they will estimate the mass screening and treatment (MSAT), vector control measures (LLIN and/or IRS) and community mobilization and participation. However, no detailed information was presented on how variables related to vector control measures (LLIN usage, for example) and community mobilization and participation will be measured to be included in the difference analysis models.

Da mesma forma, não está claro se os autores irão avaliar outras variáveis importantes na avaliação do controle da malária, tais como a variação anual da chuva e a flutuação da densidade vetorial na área estudada.

7. PLOS authors have the option to publish the peer review history of their article (what does this mean?). If published, this will include your full peer review and any attached files.

Reviewer #1: No

Reviewer #2: No

---

## [Author Response · Author response to Decision Letter 0]

30 Jul 2020

Response to reviewers

Response to points raised by academic editor

Answer1: The manuscript submitted, meets the PLOS ONE’s style requirements. The files have been named in the prescribed style as well. 

2. Please include additional information regarding the survey or questionnaire used in the study and ensure that you have provided sufficient details that others could replicate the analyses. For instance, if you developed a questionnaire as part of this study and it is not under a copyright more restrictive than CC-BY, please include a copy, in both the original language and English, as Supporting Information."

Answer 2: The questionnaire has been attached as “supportive information”.

3. In your Methods section, please provide additional information about the participant recruitment method and the demographic details of your participants. Please ensure you have provided sufficient details to replicate the analyses such as: a) the recruitment date range (month and year), b) a description of any inclusion/exclusion criteria applied to participant recruitment, c) descriptions of where participants will be recruited and where the research will take place. 

Answer 3: 

a) Each round of survey will be conducted within August to November, which synchronizes with the period for DAMaN implementation 

b) The inclusion/exclusion criteria have been defined and necessary edits made in the text. 

c) With regards to the location of the research we will select six districts of the Indian state of Odisha which are clearly mentioned in Figure 2 (currently modified as per the suggestions of 1st Reviewer). These will be selected in due course of time. 

The above statements have been included in the mansucript

Moreover, in ethics statement in the manuscript and in the online submission form, please provide additional information about the patient records (DAMaN project data) that will be used in the retrospective part of study. Specifically, please ensure that you have discussed whether all data will be fully anonymized before you accessed them and/or whether the IRB or ethics committee waived the requirement for informed consent. If patients provided informed written consent to have data from their medical records used in research, please include this information.'

DAMaN project data will be extracted in aggregate form and no individual-level data will be retrieved or used, therefore there is no need for de-identification of DAMaN project data. There is also no plan to access patient medical records. This has been included in the manuscript in the relevant section. However, for household survey we have included in the manuscript the standard statement for acquiring consent and deidentifying raw data 

4. PLOS ONE does not permit references to unpublished data; therefore, we request that you either include the referenced data or remove the instances of "data not shown," "unpublished results," or similar.

 Answer 4: We have cited a website supporting this statement and removed the “unpublished results”

5. We suggest you thoroughly copyedit your manuscript for language usage, spelling, and grammar. If you do not know anyone who can help you do this, you may wish to consider employing a professional scientific editing service. 

 Answer 5: Our colleague, Dr. Ambarish Dutta, has done thorough copyediting of the manuscript for language usage, spelling and grammar.

Answer6: We have made changes to our Data Availability statement, which has been described in the cover letter also 

Comments from 1st Reviewer

Figure 2 suggests that two districts are already selected from each of the three strata. Selection of the study districts in each stratum should be unbiased for which, the districts in each stratum may be listed from higher to lower API and then divided further in two subgroups of almost equal number of districts, one districts from upper half and one from the lower half in each zone may be selected. Sample size should be calculated based on lowest prevalence. This sample size may be applied to higher prevalence district in each of the zones for uniformity. Since prevalence of malaria has shown significant reduction in the years 2018 and 2019 in Odisha, sample size calculation should be based on current prevalence/incidence for representativeness and not based on 2016 data.

Answer1: “Selection of 2 DAMaN- implementing districts from each region of Odisha- Northern, Central, Southern; Six districts in total. Within each region, the first district will be selected from those having above median API and the second to be selected from below median API". 

This above text within quotes will be now included in Figure 2 that explains the sampling method and design. We are grateful to the reviewer for suggesting an extra layer of stratification (based on API) to make selection of districts really unbiased, hence, the selection is to be carried out now as per the newly suggested method.

 However, the sample size has been calculated for all the six districts together and not for individual district as the study objective is to compute pooled estimates mainly and not at district-wise estimates. Moreover, the DAMaN-implementing blocks and Sub-centres have similar transmission intensity so district or sub-district-wise sample size calculation for lower units may not be needed based on variances in incidence. In this context the rarest prevalence that is parasitaemia among pregnant/lactating women had been chosen as the basis for calculating sample size and it is assumed that this prevalence will be roughly similar in DAMaN-implementing pockets of the state. 

Additionally, the malaria prevalence data is not routinely collected by the malaria programme so cannot be the basis of sample size calculation. Consequently, the malaria case detection (monthly/yearly) rate (also known as monthly/yearly parasite incidence) which is a proxy for incidence of malaria has formed the basis for selection hotspots by DAMaN, and the year chosen by DAMaN planners was 2016. Therefore, the DAMaN Assessment exercise has also decided to select 2016 as the basis year for sampling. We would like to adhere to this methodology and not select the years 2018 and 2019 as basis years for sampling, during which period there was decline in malaria cases across the state. 

2. Introduction: Authors have stated ‘This initiative saw huge investment of public fund from the state government.’ How much per capita was invested, may be mentioned?

Answer 2: Agreed, the exact statement of expenditure has been mentioned in the manuscript 

3. “At least 1 ml of intra- venous blood specimen will be collected for malaria and haemoglobin analysis and anthropometric measurements will be recorded from all the members of the sampled household present at the time of interview.”

Considering only two years of implementation of DAMaN, what is the rationale of taking anthropometric measurements? What difference DAMaN would have made in a short span to these features, is not understood? Hence this objective may be dropped and focus should be primarily on the epidemiological impact.

Answer 3: 

We would like to include this component for the following reasons

• This anthropometric data may work as a baseline data for future impact evaluation

• We can compare with state anthropometric averages generated by DHS data and understand the situation in DAMaN pockets

4. What definition of an asymptomatic case will be used? Will temperature be taken at the time of house visit?

Answer 4: Self reported data of history of fever in the family in last two weeks before the date of survey will be collected in-order to understand the prevalence of symptomatic malaria. Temperature recording will not be carried out during the survey.

5. Where is P. malariae in the whole scheme? This species has been reported from Odisha with decent prevalence in some districts as in previous studies. [The prevalence of P. malariae in Odisha, India by Pati et. al., Tropical Biomedicine 34(3): 607–614 (2017)]

Answer 5: As mentioned in the manuscript (Page 16), we would assess the prevalence of P. malariae in this study. The diagnosis will be done by using species specific Polymerase Chain Reaction (PCR) as described in the methodology. 

6. The rationale for using three diagnostic methods may be clearly stated in view of Pfhrp-2 gene deletion which ranges from 5-10%; e.g. what is missed by RDT could be captured by microscopy and the PCR will capture low/sub-microscopic parasitaemia which the above two methods may miss detection.

Answer 6: Agreed to the comment. As suggested the rational behind using three diagnostic methods, namely Rapid Diagnostic Test (RDT), blood smear microscopy, polymerase chain reaction (PCR) will be used to find the prevelance of malarial parasite in the blood specimen collected. This modus-operandi will ensure an exhaustive investigation for detecting parasitaemia. The Histidine-Rich Protein 2 (PfHRP2) gene deletion in Plasmodium falciparum leads to no-detection in RDT, however Pf prevalence in those cases can be captured using microscopy and PCR. The PCR method will also be helpful in detecting parasite at sub-microscopic level, which usually gets missed out by RDT and Microscopy. The same reason three different methods have been incorporated in the text, as suggested. 

7. On page 15, objective (d), Remove `prevalence’ to assess the outcome of maternal and child health.

Answer 7: Done. “Prevalence” removed. 

8. On page 15, explain outcome assessment from MSAT? (Use Primary data/Generate data)

Answer8: Coverage of MSAT, a component of DAMaN will be estimated along with MSAT’s test positivity rate, proportion of asymptomatics among positives etc. This will be done from DAMaN project data that is secondary data.

9. On page 16, Aim 3: Explain about stratification in detail in methodology.

Answer 9: Sub-centres will be stratified into tertiles based on their DAMaN MSAT coverage and then interrupted time series analysis and difference-in-difference analyses will be carried out in these three strata based on the hypothesis that greater coverage of MSAT will lead to greater decline of malaria. Now expanded in manuscript. 

10. On page 17, Sample size: Based on the calculated sample size, you should include 2228 household with pregnant women not household only.

Answer 10: Agreed and that will be done 

11. On page 22, Statistical analysis: Anaemia should be defined separately for pregnant women, children and general population in a tabular form. It is not clear.

Answer 11: Thanks again for this very valuable suggestion. Done in the manuscript

12. Authors will need to calculate sample size separately for LLIN usage (No. of Household) and child health outcome (<5 years)

Answer 12: LLIN penetration and usage are reasonably high in these DAMaN areas from current anecdotal information. The current sample will suffice to estimate the LLIN-related proportions with enough precision. 

The same logic applies for under-5 children as this large sample size is adequate enough to estimate precise childhood health outcome given further that these health outcomes (such as undernutrition and anaemia) are not at all rare, rather rampant in these hotspots. Moreover, almost every household in these areas will have one under-5 given the current fertility pattern of this population. So, the current sampled household will suffice all the purposes. 

Review from 2nd reviewer

In the main objectives of the DAMaN Assessment the authors stated that they will estimate the mass screening and treatment (MSAT), vector control measures (LLIN and/or IRS) and community mobilization and participation. However, no detailed information was presented on how variables related to vector control measures (LLIN usage, for example) and community mobilization and participation will be measured to be included in the difference analysis models.

Answer: We are extremely grateful to the kind and encouraging words of 2nd reviewer. 

In answer to the query these indicators that were mentioned will be estimated using descriptive statistics initially. However, there variations may be modelled against probable explanatory variables, post hoc, if any interesting findings emerge from the descriptive tables.

---

## [Decision Letter · Decision Letter 1]

14 Aug 2020

Assessment of effectiveness of DAMaN : A malaria intervention program initiated by Government of Odisha, India

PONE-D-20-12484R1

Dear Dr. Ranjit,

We’re pleased to inform you that your manuscript has been judged scientifically suitable for publication and will be formally accepted for publication once it meets all outstanding technical requirements.

Kind regards,

Luzia Helena Carvalho, Ph.D.

Academic Editor

PLOS ONE

Additional Editor Comments (optional):

Reviewers' comments:

Reviewer's Responses to Questions

**Comments to the Author**

1. Does the manuscript provide a valid rationale for the proposed study, with clearly identified and justified research questions?

Reviewer #2: Yes

2. Is the protocol technically sound and planned in a manner that will lead to a meaningful outcome and allow testing the stated hypotheses?

Reviewer #2: Yes

3. Is the methodology feasible and described in sufficient detail to allow the work to be replicable?

Reviewer #2: Yes

4. Have the authors described where all data underlying the findings will be made available when the study is complete?

Reviewer #2: Yes

5. Is the manuscript presented in an intelligible fashion and written in standard English?

Reviewer #2: Yes

6. Review Comments to the Author

You may also provide optional suggestions and comments to authors that they might find helpful in planning their study.

Reviewer #2: All my comments and questions made in the first revision of this manuscript were properly argumented or answered by the authors.

7. PLOS authors have the option to publish the peer review history of their article (what does this mean?). If published, this will include your full peer review and any attached files.

Reviewer #2: No

---

## [Editor Report · Acceptance letter]

19 Aug 2020

PONE-D-20-12484R1 

Assessment of Effectiveness of DAMaN: A Malaria Intervention Program Initiated by Government of Odisha, India 

Dear Dr. Ranjit:

I'm pleased to inform you that your manuscript has been deemed suitable for publication in PLOS ONE. Congratulations! Your manuscript is now with our production department. 

Kind regards, 

on behalf of

Dr. Luzia Helena Carvalho 

Academic Editor

PLOS ONE